# Effect of Juvenile Hormone on Resistance against Entomopathogenic Fungus *Metarhizium robertsii* Differs between Sexes

**DOI:** 10.3390/jof6040298

**Published:** 2020-11-19

**Authors:** Markus J. Rantala, Ivan M. Dubovskiy, Mari Pölkki, Tatjana Krama, Jorge Contreras-Garduño, Indrikis A. Krams

**Affiliations:** 1Department of Biology, Section of Ecology, University of Turku, FI-20014 Turku, Finland; mjranta@utu.fi (M.J.R.); mari.polkki@hotmail.com (M.P.); 2Laboratory of Biological Plant Protection and Biotechnology, Department Plant Protection, Novosibirsk State Agrarian University, 630039 Novosibirsk, Russia; dubovskiy2000@yahoo.com; 3Siberian Federal Scientific Centre of Agro-BioTechnologies, Russian Academy of Sciences, 630501 Krasnoobsk, Russia; 4Department of Biotechnology, Daugavpils University, 5401 Daugavpils, Latvia; tatjana.krama@du.lv; 5Department of Plant Health, Estonian University of Life Sciences, 51006 Tartu, Estonia; 6Laboratorio de Ecología Evolutiva, Escuela Nacional de Estudios Superiores, Universidad Nacional Autónoma de México, Mexico City 04510, Mexico; jcg@enesmorelia.unam.mx; 7Institute of Ecology and Earth Sciences, University of Tartu, 51014 Tartu, Estonia; 8Department of Zoology and Animal Ecology, Faculty of Biology, University of Latvia, 1004 Rīga, Latvia

**Keywords:** immune defense, immunocompetence, *Metarhizium robertsii*, pathogens, sex, *Tenebrio molitor*

## Abstract

Juvenile hormone has been suggested to be a potential mediator in the trade-off between mating and insects’ immunity. Studies on various insect taxons have found that juvenile hormone interferes with humoral and cellular immunity. Although this was shown experimentally, studies using highly virulent parasites or pathogens are lacking so far. In this study, we tested if juvenile hormone administration affected resistance against entomopathogenic fungi, *Metarhizium robertsii*, in the mealworm beetle, *Tenebrio molitor*. In previous studies with *T. molitor,* juvenile hormone has been found to reduce a major humoral immune effector-system (phenoloxidase) in both sexes and decrease the encapsulation response in males. Here, we found that juvenile hormone administration prolonged survival time after infection with *M. robertsii* in males but reduced survival time in females. This study indicates that the effects of juvenile hormone on insect immunity might be more complicated than previously considered. We also suggest that there might be a trade-off between specific and non-specific immunity since, in males, juvenile hormone enhances specific immunity but corrupts non-specific immunity. Our study highlights the importance of using real parasites and pathogens in immuno-ecological studies.

## 1. Introduction

The immunocompetence handicap hypothesis [1] suggests that the expression of secondary sexual traits honestly signals male quality because testosterone—needed to develop these traits—has immunosuppressive effects. However, many studies in vertebrates have failed to reveal a clear relationship between the expression of secondary sexual ornamentation and immune defense. This might be because the physiological relationship between these traits is not as simple as originally thought and because of the alterations of stress hormones, which may involve trade-offs between sexual ornamentation and immunity [2]. However, many studies in insects and spiders have found that the expression of males’ secondary sexual characteristics correlates positively with their immune defense [3,4,5,6,7]. Since insects lack male-specific hormones such as testosterone, it has been suggested that in these animals, the immunocompetence handicap mechanism would be mediated by juvenile hormone (JH) [8].

Juvenile hormone is synthesized in the corpora allata, and it is known to play a crucial role in many aspects of development, reproduction, aging, and behavior in insects. For example, it seems that juvenile hormone type III is associated with sex pheromone production in cockroaches [9]. Similarly, it was found that the administration of juvenile hormone increased the attractiveness of male pheromones in the mealworm beetle, *Tenebrio molitor*. Still, it reduced the strength of the encapsulation response and phenoloxidase activity of hemolymph [8]. Interestingly, it has been found that while JH increases male attractiveness, it reduces the size of the testis and sperm viability, suggesting another cost of high juvenile hormone levels [10]. In the territorial damselfly, *Calopteryx virgo*, it has been found that the administration of methoprene acid (an analog of JH) increased aggression and occupation time in territories but decreased phenoloxidase activity of hemolymph [11]. Juvenile hormone has also been shown to affect genes related to antibacterial peptide expression [12,13]. In the diamondback moth, *Plutella xylostella*, it was shown that juvenile hormone inhibited hemocyte-spreading behavior [13], suggesting that JH induces immune suppression because effective hemocyte-spreading is important for phagocytosis, nodulation, and encapsulation [14]. Furthermore, they found that pyriproxyfen (a JH analog) enhanced the pathogenicity of *Bacillus thuringiensis* subsp. *kurstaki* [13].

Juvenile hormone may also be associated with a trade-off between reproduction and immunity in insects. For example, mating reduced the activity of phenoloxidase enzyme in the hemolymph of both sexes of *T. molitor* [15], but there was no effect on the hemocyte load. It has been found that the observed decrease in phenoloxidase levels was caused by an increase in juvenile hormone levels due to mating, which indicates that juvenile hormone might indeed mediate the trade-off between mating and immunity [15]. Previously, it was shown that mating enhances resistance against entomopathogenic fungi, *Beauveria bassiana,* infection and that the effect was stronger in males than among females of *T. molitor* [16]. This shows that the effect of juvenile hormone on the immune system may be more complicated than previously thought. Overall, these results suggest that studies testing the effect of juvenile hormone on the resistance against highly virulent parasites and pathogens are needed.

Entomopathogenic fungi (EPF), including entomophthoralean fungi, offer environmentally friendly alternatives to conventional synthetic chemicals for arthropod pest control. There are over 750 different species of EPF identified so far [17]. Although entomophthoralean fungi are highly efficacious, much attention has focused on researching species belonging to the order Hypocreales because they are more amenable for mass production and have a relatively wide host range. Approximately 80% of the commercially available EPF products are based on the *Metarhizium robertsii, M. anisopliae, M. brunneum,* and *B. bassiana*.

The study aimed to test whether juvenile hormone affects the resistance of *T. molitor* against a real pathogen and whether there is a sex difference in the pathogen resistance. We tested juvenile hormone-related effects on the immunity of *T. molitor* against *M. robertsii*.

## 2. Materials and Methods

### 2.1. Study Animals

*T. molitor* beetles used in the experiment originated from a natural population. They were collected from several barns in southeastern Latvia in 2007 [18] and maintained at the University of Turku. They were reared in plastic boxes (5 l) and fed with wheat bran and apple at a constant temperature of 28 °C under a 14 L:10 D photoperiod and constant humidity of 70%. We collected pupae daily and determined the sex of each pupa by examining the developing genitalia on the eighth abdominal segment. Shortly after emergence, the beetles were placed individually in plastic film roll canisters with an excess of fresh apple. Sexes were physically isolated to ensure virginity. We excluded individuals that had visible developmental abnormalities or whose size deviated strongly from the population mean. Beetles of each sex were randomly allocated to the treatments when aged between 10 and 14 days. Before the experiments, we weighed the fresh body mass of each beetle to the nearest 0.1 mg.

### 2.2. Experimental Treatments

In total, we had 242 females and 171 males. The insects were randomly allocated to each treatment in which beetles were injected ventrally either 5 or 10 μg of JH type III (Sigma, St Quentin Fallavier, France) in 5 μL of Ringer: acetone (9:1) solution between the 2nd and 3rd sternite region using a 10 μL Hamilton syringe (30 G) (Hamilton Company, Switzerland). Control males and females received 5 μL of Ringer: acetone solution only, but otherwise, both groups were treated identically. Before injection, the beetles were anesthetized with carbon dioxide. The inoculated beetles were placed individually into plastic film roll canisters kept in an environmental chamber at 24 °C under a 16 L:8 D photoperiod and fed with fresh apple.

During the following day, we infected the beetles with the conidia of entomopathogenic fungi, *M. robertsii* [19]. To infect the insects with the fungi, we anesthetized beetles with carbon dioxide. We dropped dorsally 5 µL of LD50 solution containing conidium (5 × 106 conidium/mL) with a pipette on the abdomen under the beetles’ wings. LD50 doses were determined in preliminary experiments by infecting a separate group of insects with different doses of conidium and selecting the dose that closest to kill 50% of the treated animals. Unfortunately, the mortality caused by the fungi was much higher in the experiment than in our preliminary studies (probably because wounding by the needle made it easier for the fungi to penetrate the cuticula). In our previous studies on *T. molitor*, we found that juvenile hormone administration did not influence on beetles’ survival [8]. There was no mortality among beetles when dipped in the control solution. Thus, we left the control solution out from this experiment to double the sample size in the fungal treatment groups. After the infection with fungi, beetles were placed individually back to the plastic film roll canisters in an environmental chamber at 24 °C under a 16 L:8 D photoperiod and fed with fresh apple for 21 days to check daily for mortality rates of experimental individuals.

### 2.3. Statistical Analyses

We used Cox proportional hazards regression (survival analysis) to examine survival differences between the different treatments after infection with the entomopathogenic fungi. In the model, we presented sex and JH hormone treatment as categorical covariates, mass as a continuous covariate, and survival time as a dependent variable. We initiated a model fitting with a model that included all the main effects and the two-way interaction terms that best address the subject of interest. We searched for the best model using a backward stepwise method (backward LR). We conducted all statistical analyses by using PASW Statistic 18.0 for Windows.

## 3. Results

The best model predicting survival after the fungal infection contained treatment (Wald = 5.520, df = 2, *p* = 0.063), 5 μg of JH treatment (OR = 1.030, Wald = 0.025, df = 1, *p* = 0.875), 10 μg of JH treatment (OR = 0.693, Wald = 3.848, df = 1, *p* = 0.050), sex × treatment (Wald = 10,706, df = 2, *p* = 0.005), sex × 5 μg of JH treatment (OR = 1.099, Wald = 0.025, df = 1, *p* = 0.875) and sex × 10 μg of JH treatment (OR = 2135, Wald = 9136, df = 1, *p* = 0.003). The females had stronger resistance against the fungi than males in the control treatment group (Wald = 3.897, *p* = 0.048). However, administration of the small dose (5 μg) of juvenile hormone did not have any effect on the survival after the fungal infection in either of the sexes. Instead, administration of the larger juvenile hormone dose (10 μg) increased male survival but decreased females survival after the fungal infection (Figure 1a,b). Thus, it seems that juvenile hormone enhances male’s immunity but corrupts female immunity.

## 4. Discussion

In this study, we found that juvenile hormone enhances resistance against the entomopathogenic fungi in the males of *T. molitor*, which contradicts the results by some previous studies [8,11,15], which have found that juvenile hormone corrupts immunity. However, we found that juvenile hormone reduced resistance against the entomopathogenic fungi in females. Thus, the effects of juvenile hormone on the immune system are much more complicated than previously thought. The reason why juvenile hormone had a different effect on the immune system in males and females in our study remains unclear. One possibility is that instead of corrupting the immunocompetence, juvenile hormone may rather cause a reallocation of the resources to those parts of the immune defense which need them the most (as has been proposed in vertebrates for testosterone and immunity [20]). Since optimal life history strategies differ between the sexes, the optimal reallocation of the resources between different immunity arms may also differ.

Interestingly, there were sex differences in resistance in the control treatment group: females having stronger resistance against the fungi than males. It has been suggested that the ultimate mechanism for the observed sex differences in immune function could be a differential selection favoring different investment levels in the immune defense system [21,22]. Because female fitness is limited by the number of offspring produced, whereas male fitness is limited by the number of females fertilized, males are expected to invest more in sexual competition and current reproduction at the expense of their immune defense compared to females (the Bateman Principle) [21]. However, an experimental study found no sex differences in parasite infections among arthropod hosts [23]. Likewise, efforts to examine the sex differences in innate immune function in insects have been met with mixed results [24]. Thus, more studies testing the sex differences in insects using real parasites and pathogens would be needed to test the Bateman principle in insects. On the other hand, it was shown that sex-specific responses to experimental manipulation of fitness-limiting resources affects both the magnitude and direction of sex differences in immune function [22,25]. This suggests that for species similarly limited in their reproduction, phenotypic plasticity would be an important determinant of sex differences in immune function and other life-history traits. Likewise, immunological sex differences were found in the autumnal moth, *Epirrita autumnata*, which varied in populations differing in their degree of inbreeding [26]. Thus, it seems that there are plausible explanations for sexual dimorphism in immunity other than just the Bateman principle, which is traditionally used to explain the observed sex difference in immunity [21].

Since our previous studies with *T. molitor* found that the administration of juvenile hormone reduced phenoloxidase activity and the encapsulation response against a nylon monofilament [8], the results of this study suggest that the effect of juvenile hormone differs between specific and non-specific immunity in *T. molitor*. Interestingly, in the autumnal moth, *E. autumnata*, the encapsulation response against a nylon monofilament was positively associated with the resistance against *B. bassiana* [26]. However, it has been shown that cellular antifungal reactions, such as phagocytosis and multicellular encapsulations, are suppressed during the development of fungal diseases [27]. Thus, encapsulation or phenoloxidase activities may not mirror the resistance against fungal pathogens, being indirectly correlated via the individuals’ general condition. Thus, it seems that the association between the specific and non-specific parts of immunity appears to be very complicated. Our study highlights the importance of using real parasites and pathogens in immuno-ecological studies.

## Figures and Tables

**Figure 1 jof-06-00298-f001:**
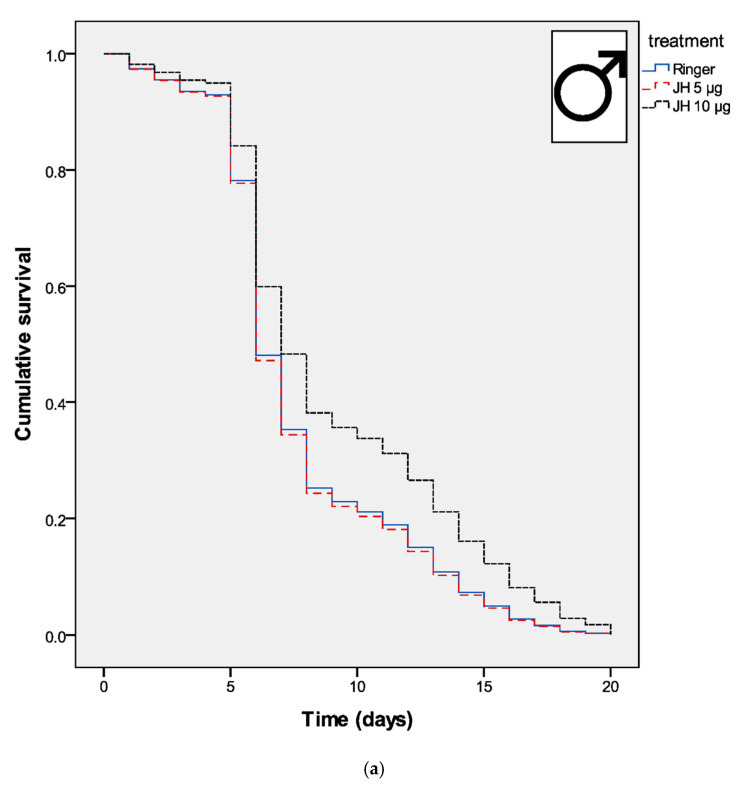
Cumulative survival of (**a**) male and (**b**) female beetles after topical application with the entomopathogenic fungi *M. robertsii*. Curves represent the survival functions calculated by the Cox regression survival analysis.

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
