# Peer review of "Effect of Juvenile Hormone on Resistance against Entomopathogenic Fungus Metarhizium robertsii Differs between Sexes"

_jof, 2020, doi:10.3390/jof6040298_

Round 1
Reviewer 1 Report
The authors describe their findings about sex-biased survival against entomopathogenic fungi in T. molitor, and how this can be influenced by juvenile hormone. Although their conclusions are based on a single experiment, the findings are interesting to the field. Minor revisions are needed:
-Citations should be numbered in the text. Please read carefully the "authors guidelines".
-Line 86: Please indicate the species of beetles used
-Line 89 and 103 and 117: Be consistent when idicating temperature in Celsius (e.g. 28°C, not +28C°)
-Line 100: "Control males received 5 uL of Ringer solution..." and what received females?
-Line 107: Can you be more specific about the site of treatment on the insect? LD50 solution was applied dorsally or ventrally?
-Statistical analysis and Results: How many pairwise comparisons did you performed? Did you use any correction (e.g. Bonferroni, FDR)?
-Figure: Please specify how are insect treated in the assay (e.g. injections, dippin, topical application)
- Line 157: You stated that, "in the control group, there was sex differences in resistance". However is not clear, from results, where are the significant differences described (i.e. statistics describing the differences among controls), supporting your sentence.
- Line 168: I wouldn't say that a 15 years ago study (McKeab and Nunney, 2005) it's a recent study.
- Please check references and correct the caps-lock style used for many journal names.
Author Response
REVIEWER 1
The authors describe their findings about sex-biased survival against entomopathogenic fungi in T. molitor, and how this can be influenced by juvenile hormone. Although their conclusions are based on a single experiment, the findings are interesting to the field. Minor revisions are needed:
We are thankful to the reviewer for pointing out our mistakes, which helped to improve the manuscript.
-Citations should be numbered in the text. Please read carefully the "authors guidelines".
RESPONSE: Citations have been numbered according to the guidelines in the revised text.
-Line 86: Please indicate the species of beetles used
RESPONSE: This is done on line 99.
-Line 89 and 103 and 117: Be consistent when idicating temperature in Celsius (e.g. 28°C, not +28C°)
RESPONSE: This is fixed on lines 98-100, 115 and 127.
-Line 100: "Control males received 5 uL of Ringer solution..." and what received females?
RESPONSE: Fixed on line 112.
-Line 107: Can you be more specific about the site of treatment on the insect? LD50 solution was applied dorsally or ventrally?
RESPONSE: Please, find more about this on line 118 (dorsally).
-Statistical analysis and Results: How many pairwise comparisons did you performed? Did you use any correction (e.g. Bonferroni, FDR)?
RESPONSE: We did not apply corrections since they did not affect the results (line 136-137).
-Figure: Please specify how are insect treated in the assay (e.g. injections, dippin, topical application)
RESPONSE: This has been improved. Please, find more information in the Figure legend.
- Line 157: You stated that, "in the control group, there was sex differences in resistance". However is not clear, from results, where are the significant differences described (i.e. statistics describing the differences among controls), supporting your sentence.
RESPONSE: This has been added to the Results on lines 146-147.
- Line 168: I wouldn't say that a 15 years ago study (McKeab and Nunney, 2005) it's a recent study.
RESPONSE: We deleted “recent study”.
- Please check references and correct the caps-lock style used for many journal names.
RESPONSE: Many thanks for reminding us about this important point! Done according the requirements of the journal.
Reviewer 2 Report
The study titled “Effect of juvenile hormone on resistance against entomopathogenic fungus Metharizium robertsii differs between sexes” tests the phenomenon that juvenile hormone administration reduced the immunity against fungal pathogens. The authors determined the mortality of male and female adults at two doses of JH and an LD50 of entomopathogen fungus (M. robertii). Males showed an improved response to the fungus relative to the females although females had inherent ability to be resistant against the entomopathogen. Overall, it is an interesting study and the results support the hypothesis.
The introduction was clearly written with sufficient background and the study was conducted with sufficient number of insects. However, it was not clear if a single insect is considered as a biological replicate. There were several grammatical errors, wrong word choice and typos throughout the manuscript.
Major comments: The rationale of the study was not mentioned clearly. The definition of the real pathogens mentioned by the authors is not clear. Although no mortality was observed at 5ug of JH, the authors did not perform any other tests to confirm the absence of any cellular responses of the immune system. Accounting for only survivorship of the experimental insects can lead to misleading conclusions and is not comprehensive. Also, the study seems sike an addendum to the study already published and cannot be accepted as a full article. It could possibly be a short communication. The authors could include more than one pathogen type to study the effects of juvenile hormone on the immunity of insects.
Specific comments:
Line 21: Please delete “sexual signaling and immune defense” as the latter part of the sentence is explaining the same.
Line 22: Replace “corrupt” with interfere
Line 23: Rephrase the rationale of the study. This sentence does not relate to the previous sentence.
Line 38: replace “honest signals” if it was used to explain the hypothesis or if it is a part of the hypothesis, please define it.
Line 79: What is the definition of real parasites and pathogens?
Line 83: What is the rationale for choosing M. robertsii?
Line 87: Delete “by the authors”
Line 88: Delete “bulk” instead mention the dimensions of the vials.
Line 89: What is the relative humidity and photoperiod conditions of growth?
Line 98: Delete semicolon and “in which”
Line 103: 24oC
Line 109: “treated insects”?
Line 109: Was JH injected at the same place as the conidia inoculation?
Line 113: This sentence is confusing. Do the authors mean to say that for the preliminary experiments the insects were dipped in conidia to determine the LD50?
Line 117: 24oC
Line 194: References need format changes.
Author Response
REVIEWER 2
The study titled “Effect of juvenile hormone on resistance against entomopathogenic fungus Metharizium robertsii differs between sexes” tests the phenomenon that juvenile hormone administration reduced the immunity against fungal pathogens. The authors determined the mortality of male and female adults at two doses of JH and an LD50 of entomopathogen fungus (M. robertii). Males showed an improved response to the fungus relative to the females although females had inherent ability to be resistant against the entomopathogen. Overall, it is an interesting study and the results support the hypothesis.
The introduction was clearly written with sufficient background and the study was conducted with sufficient number of insects.
We are thankful to the reviewer for pointing out our mistakes, which helped to improve the manuscript.
However, it was not clear if a single insect is considered as a biological replicate.
RESPONSE: Sure, because death is an individual phenomenon.
There were several grammatical errors, wrong word choice and typos throughout the manuscript.
RESPONSE: This is fixed. We hope that English has been substantially improved in the revised version of the manuscript.
Major comments: The rationale of the study was not mentioned clearly. The definition of the real pathogens mentioned by the authors is not clear. Although no mortality was observed at 5ug of JH, the authors did not perform any other tests to confirm the absence of any cellular responses of the immune system. Accounting for only survivorship of the experimental insects can lead to misleading conclusions and is not comprehensive. Also, the study seems sike an addendum to the study already published and cannot be accepted as a full article. It could possibly be a short communication. The authors could include more than one pathogen type to study the effects of juvenile hormone on the immunity of insects.
RESPONSE: We tried to make the text more focused and also tried to explain our experimental approach in more detail on lines 116-128.
If the reviewer and/or editor insist, we can transform the MS into a Short Communication paper.
Specific comments:
Line 21: Please delete “sexual signaling and immune defense” as the latter part of the sentence is explaining the same.
RESPONSE: We deleted “sexual signaling and immune defense and also between”.
Line 22: Replace “corrupt” with interfere
RESPONSE: Replaced with “interferes with” (line 28).
Line 23: Rephrase the rationale of the study. This sentence does not relate to the previous sentence.
RESPONSE: We changed the sentence into the following form: “Although this was shown experimentally, studies using highly virulent parasites or pathogens are lacking so far” (lines 29-30).
Line 38: replace “honest signals” if it was used to explain the hypothesis or if it is a part of the hypothesis, please define it.
RESPONSE: We improved the sentence by replacing “honest signals” with “honestly signals” (line 48).
Line 79: What is the definition of real parasites and pathogens?
RESPONSE: “real” replaced with “highly virulent” (line 90).
Line 83: What is the rationale for choosing M. robertsii?
RESPONSE: Information about entomopathogenic fungi has been added to the Introduction (lines 82-88)
Line 87: Delete “by the authors”
RESPONSE: Done as suggested by the reviewer.
Line 88: Delete “bulk” instead mention the dimensions of the vials.
RESPONSE: Fixed on line 99.
Line 89: What is the relative humidity and photoperiod conditions of growth?
RESPONSE: This has been added on line 100.
Line 98: Delete semicolon and “in which”
RESPONSE: Done.
Line 103: 24oC
RESPONSE: Fixed.
Line 109: “treated insects”?
RESPONSE: We deleted “non-” (line 113).
Line 109: Was JH injected at the same place as the conidia inoculation?
RESPONSE: It was different parts of the insects. We updated this on lines 110 and 118.
Line 113: This sentence is confusing. Do the authors mean to say that for the preliminary experiments the insects were dipped in conidia to determine the LD50?
RESPONSE: This part has been improved on lines 116-128.
Line 117: 24oC
RESPONSE: Fixed.
Line 194: References need format changes.
RESPONSE: Done according the requirements of the journal.